# Genome-wide analysis highlights genetic admixture in exotic germplasm resources of *Eucalyptus* and unexpected ancestral genomic composition of interspecific hybrids

**Danyllo Amaral de Oliveira**[1], **Paulo Henrique Muller da Silva**[2], **Evandro Novaes**[1], **Dario Grattapaglia**[3]*

1 Departamento de Biologia, Universidade Federal de Lavras, Lavras, MG, Brazil, 2 Instituto de Pesquisas e Estudos Florestais, Piracicaba, SP, Brazil, 3 Plant Genetics Laboratory, EMBRAPA Genetic Resources and Biotechnology, Brasilia, DF, Brazil

* dario.grattapaglia@embrapa.br

**Data Availability Statement:** All relevant data are within the paper and its Supporting Information files.

## Abstract

*Eucalyptus* is an economically important genus comprising more than 890 species in different subgenera and sections. Approximately twenty species of subgenus *Symphyomyrtus* account for 95% of the world's planted eucalypts. Discrimination of closely related eucalypt taxa is challenging, consistent with their recent phylogenetic divergence and occasional hybridization in nature. Admixture, misclassification or mislabeling of *Eucalyptus* germplasm resources maintained as exotics have been suggested, although no reports are available. Moreover, hybrids with increased productivity and traits complementarity are planted worldwide, but little is known about their actual genomic ancestry. In this study we examined a set of 440 trees of 16 different *Eucalyptus* species and 44 interspecific hybrids of multi-species origin conserved in germplasm banks in Brazil. We used genome-wide SNP data to evaluate the agreement between the alleged phylogenetic classification of species and provenances as registered in their historical records, and their observed genetic clustering derived from SNP data. Genetic structure analyses correctly assigned each of the 16 species to a different cluster although the PCA positioning of *E. longirostrata* was inconsistent with its current taxonomy. Admixture was present for closely related species' materials derived from local germplasm banks, indicating unintended hybridization following germplasm introduction. Provenances could be discriminated for some species, indicating that SNP-based discrimination was directly proportional to geographical distance, consistent with an isolation-by-distance model. SNP-based genomic ancestry analysis showed that the majority of the hybrids displayed realized genomic composition deviating from the expected ones based on their pedigree records, consistent with admixture in their parents and pervasive genome-wide directional selection toward the fast-growing *E. grandis* genome. SNP data in support of tree breeding provide precise germplasm identity verification, and allow breeders to objectively recognize the actual ancestral origin of superior hybrids to more realistically guide the program toward the development of the desired genetic combinations.

**Funding:** This work was supported by FAPESP (Foundation for Scientific Research of São Paulo State) competitive grant number 2017/24609-5 to PHMS, FAP-DF (Foundation for Scientific Research of the Federal District) competitive grant RECGENOMICS 00193-00000924/2021-92 to DG and CNPq (Brazilian National Council for Scientific and Technological Development) research productivity fellowship to PHMS, EN and DG. There was no additional external funding received for this study and the funders had no role in study design, data collection and analysis, decision to publish, or preparation of the manuscript.

**Competing interests:** The authors have declared that no competing interests exist.

## Introduction

*Eucalyptus* is a highly diverse genus of tree species from Australia and neighboring islands ruling the forest landscape across large part of Australia and neighboring islands. Besides its keystone ecological role in native forests, the genus includes the most widely commercially planted hardwood tree species in tropical and subtropical regions of the world [1, 2]. Fast growth, wide adaptability to a globally broad diversity of tropical and subtropical environments, combined with multipurpose wood properties for energy, solid wood products, pulp and paper, have secured the superior position of the eucalypts in current world forestry [3].

As particularly speciose, the genus *Eucalyptus* has received significant attention to its phylogenetic organization. The classic taxonomy of eucalypts with more than 700 species [4], has now been expanded to include over 890 species [5], but taxonomic issues still remain as species delimitations are still being actively investigated. The genus was originally subdivided in 13 subgenera but two of them, *Angophora* and *Corymbia*, were recently recognized as separate genera based on molecular evidences [6, 7]. The largest subgenus *Symphyomyrtus* includes 470 species organized in 15 sections which are mostly well separated using molecular marker data [8, 9]. Nevertheless, some discrepancies exist between DNA marker and morphology-based classifications at the lower taxonomic level, consistent with recent divergence of taxa, characters convergence and occasional hybridization in natural populations, generating hybrid swarms and zones of intergradation between species of the same section in the wild [10–13].

Most of the economically important eucalypts species belong to subgenus *Symphyomyrtus*, and, within it, to three specific sections, *Exsertaria*, *Latoangulatae* and *Maidenaria* [14]. These sections include approximately twenty species that account for more than 95% of the world's planted eucalypts. Among those, *Eucalyptus grandis*, *E. urophylla* (sect. *Latoangulatae*), *E. camaldulensis* (sect. *Exsertaria*) and *E. globulus* (sect. *Maidenaria*) make up more than 80% of the planted areas [15]. The sexual compatibility among species within and across these three main sections have been important drivers of breeding programs especially in tropical and subtropical countries, where large extensions of forests are planted with hybrid material [10, 12, 16, 17]. Hybrid breeding coupled to clonal propagation has allowed the aggregation and exploitation of important characteristics from different species and provenances in highly productive hybrid clones [18, 19]. In Brazil, an estimated 80% of eucalypts plantations are established with first- or second-generation hybrids involving mainly *E. urophylla* and *E. grandis* [17, 20]. The remaining 20% are mostly pure species material of the two former species, or hybrids with *E. camaldulensis* and *E. pellita* for greater drought tolerance, and to a much lesser extent with *E. globulus* to improve wood quality for cellulose. Additionally, *E. dunnii* and *E. benthamii* are also used as pure species or in hybrid combinations in areas subject to frost [21].

A number of studies in the last fifteen years have approached and largely established the challenge of resolving lower-level, within section taxonomy in *Eucalyptus* using different genome-wide DNA marker data [8, 9, 22–24]. However, issues remain for species in section *Latoangulatae*, for example, due to their intermediate nature, when compared to more densely clustered taxa in other sections [8]. Admixture of *Eucalyptus* species in their native range has been reported [11, 25], reflecting the phylogenetic fluidity that still exist in some taxa. However, misclassification, mislabeling and hybridization of eucalypts germplasm resources maintained as exotics in different countries has been suggested, but reports are only anecdotal. Little or no data exist on such incidences, or on the overall current status of gene banks in countries where eucalypt make up a large proportion of planted forests.

Genome-wide studies looking at large numbers of eucalypt species have used DArT markers, genotyped originally with probe arrays [26] and, more recently, by genome complexity reduction with restriction enzyme digestion followed by high-throughput sequencing [27].

This DNA assay has been valuable as it provides simultaneous discovery and genotyping of Single Nucleotide Polymorphisms (SNP) within and across species, facilitating genus-wide phylogenetic studies. However, some challenges remain for this SNP genotyping method due to variable sequencing coverage and irregular sampling of loci causing variable genotype reproducibility and ultimately limited data portability across studies in highly heterozygous genomes such as those of the eucalypts [28–30]. The development of *Eucalyptus* multispecies SNP arrays based on industry-level "gold standard" technology has provided a worldwide usable platform allowing seamless and precise data exchange across studies [31].

Although species discrimination using DNA data is largely settled, less attention has been devoted to looking at provenance variation within species. This is particularly important for breeding programs that take advantage of matching distinctive provenance characteristics to specific sites in exotic environments, or aim at deliberately exploiting provenance and species complementarity by building specific genomic compositions by interspecific hybridization [2, 12, 18]. Likewise, few studies have examined the possibility of using DNA data to describe the actual genomic ancestral composition of hybrids, including those derived from more than two parental species. Knowledge of the actual genomic composition of complex hybrids of distinctive performance would allow directing more deliberate selection strategies in hybrid breeding programs. Earlier studies using microsatellite markers indicated that provenances of *E. grandis* could be distinguished but not for *E. urophylla* and *E. camaldulensis*, and some hybrid clones could be assigned to their most likely ancestral species, although with incomplete resolution [32]. Using SNP data, preliminary analyses have shown that provenances within species could be distinguished for *E. grandis* and *E. urophylla* [31, 33] but not for *E. camaldulensis*, consistent with the latter being more prone to hybridization or a remnant of an ancient widespread taxon [8].

The current eucalypt SNP arrays have been used to estimate recombination rates and carry out dense linkage mapping [34], build relationship matrices for genomic selection in several species, reviewed in [35], and understand the consequences of artificial selection [36]. No studies to date, however, have evaluated their ability to characterize germplasm material in gene banks. Questions frequently arise regarding the verification of the alleged species classification, the possibility of discriminating provenances and determining the genomic composition of hybrid clones of unknown or uncertain origin derived from successive generations of interspecific recombination. In this study we examined a large set of germplasm accessions including 440 *Eucalyptus* trees of 16 species and 44 interspecific hybrids currently conserved or used in Brazil. We used genome-wide SNP data to evaluate the agreement between the alleged phylogenetic classification of species and provenances as registered in their historical records, and their observed genetic clustering obtained from genomic data, agnostic to any prior phylogenetic information. We focused on the main planted species of *Symphyomyrtus* given their outstanding relevance in terms of germplasm use and conservation. Additionally, we used SNP data to examine the actual genomic makeups of hybrids derived from interspecific crosses involving two of more species, and compare them with their expected composition based on the recorded ancestral species.

## Material and methods

### Plant material

The study involved a germplasm set of 440 trees belonging to 16 *Eucalyptus* species of five sections of subgenus *Symphyomyrtus* and 44 interspecific hybrid clones (Table 1). These trees are conserved in species/provenance/progeny trials and clonal banks at the Anhembi Experimental Research Station of the Institute for Forestry Research (IPEF) in Brazil (22.7897˚ S,

**Table 1. Section, provenances, source and number of individual trees sampled for each of the 16 *Eucalyptus* species included in the study.**

| Species | Section | Provenance | Source* | Number of individuals |
|---|---|---|---|---|
| *E. argophloia* | Adnataria | Narromine | CSIRO 12716 | 16 |
| *E. deglupta* | Equatoria | Philipines | CSIRO 21163 | 18 |
| *E. brassiana* | Exsertaria | Cape York Peninsula | CSIRO 1188 | 18 |
| *E. camaldulensis* | Exsertaria | Nott's Crossing Katherine River | IPEF—Brazil | 26 |
| *E. tereticornis* | Exsertaria | Unknown | Suzano—Brazil | 7 |
| *E. tereticornis* | Exsertaria | MT Garnet | CSIRO 20768 | 10 |
| *E. tereticornis* | Exsertaria | N of Mareeba | CSIRO 15370 | 12 |
| *E. tereticornis* | Exsertaria | Mitchell R Oaky CK | CSIRO 16645 | 12 |
| *E. grandis* | Latoangulata | Atherton | Suzano—Brazil | 16 |
| *E. grandis* | Latoangulata | Coffs Harbour | Suzano—Brazil | 20 |
| *E. longirostrata* | Latoangulata | Starkvale Creek | CSIRO 20007 | 12 |
| *E. longirostrata* | Latoangulata | Coominglah SF | CSIRO 19312 | 12 |
| *E. longirostrata* | Latoangulata | Goodger | CSIRO 20943 | 12 |
| *E. pellita* | Latoangulata | West of Hopevale | CSIRO 21018 | 16 |
| *E. pellita* | Latoangulata | NW of Kuranda | CSIRO 17859 | 15 |
| *E. robusta* | Latoangulata | David Low Wy Brisbane | CSIRO 17004 | 16 |
| *E. robusta* | Latoangulata | Byfield SF | CSIRO 15945 | 14 |
| *E. saligna* | Latoangulata | 20 km N Helidon | CSIRO 16942 | 12 |
| *E. saligna* | Latoangulata | Kroombit Tops | CSIRO 20483 | 12 |
| *E. saligna* | Latoangulata | Richmond Range | CSIRO 20483 | 12 |
| *E. urophylla* | Latoangulata | Multiple provenances | IPEF—Brazil | 42 |
| *E. benthamii* | Maidenaria | Kedumba Valley | Klabin- Brazil | 24 |
| *E. dunnii* | Maidenaria | Unknown | CMPC—Brazil | 28 |
| *E. globulus* | Maidenaria | Unknown | CMPC—Brazil | 26 |
| *E. nitens* | Maidenaria | Unknown | Klabin—Brazil | 20 |
| *E. viminalis* | Maidenaria | Unknown | Klabin—Brazil | 12 |
| Total | | | | 440 |

* The CSIRO seed lots were sampled in native populations from Australia. The Brazilian germplasm, maintained as exotic resources, are indicated by the institution or forest company where the germplasm is conserved or utilized for breeding. These Brazilian germplasm resources are at least one generation removed from the original seedlots introductions.

48.1280˚ W), or in the gene banks of some associated forest-based companies. For six species (*E. grandis*, *E. longirostrata*, *E. pellita*, *E. robusta*, *E. saligna* and *E. tereticornis*), samples were analyzed for more than one provenance. The original locations of the species and provenances were plotted on top of a base map of world country boundaries shapefile of Australia, publicly available under a Creative Commons Attribution 4.0 International Public License (https:// datacatalog.worldbank.org/search/dataset/0038272/World-Bank-Official-Boundaries) using the R package tmap (Fig 1). Plant material included: (1) individual trees sampled in species/ provenance trials established with original seeds collected in Australia for which the CSIRO (Commonwealth Scientific and Industrial Research Organisation) seedlot number is known, and (2) individual trees collected in Brazilian germplasm banks at least one generation removed from the original introductions, maintained by IPEF or by three associated forestry companies (Suzano, Klabin, Vallourec and CMPC Celulose Riograndense), sometimes with unknown provenance origin (Table 1). In addition, 44 interspecific hybrid clones obtained by controlled interspecific crosses of two or more species were studied to compare their SNP-realized versus pedigree-expected genomic composition. These hybrids were grouped into five

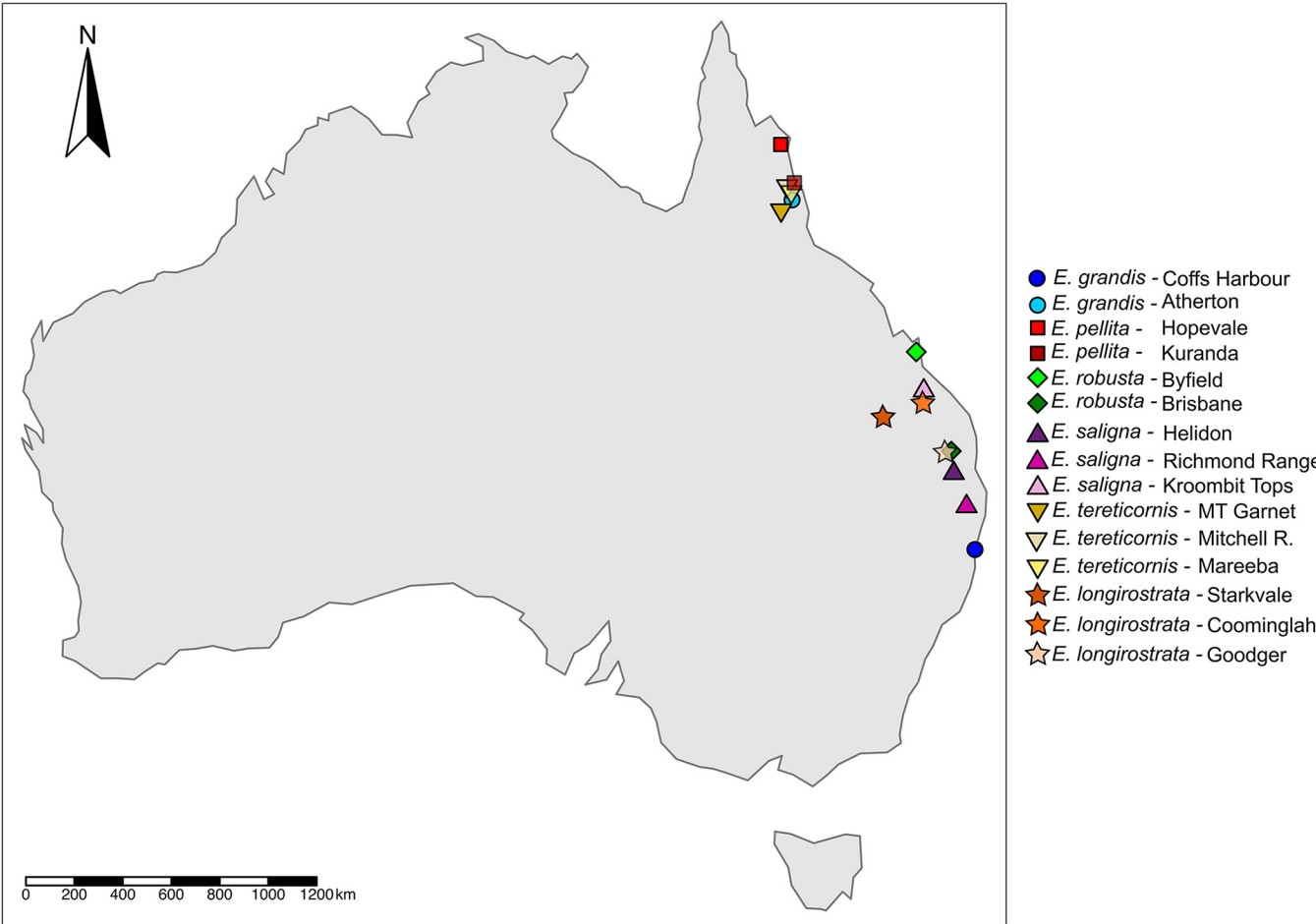

**Fig 1. *Eucalyptus*** **species for which provenances were studied, plotted on their respective geographic locations on a publicly available basemap reprinted under a CC BY 4.0 license with permission from The World Bank (https://datacatalog.worldbank.org/search/dataset/0038272/World-Bank-Official-Boundaries).**

classes (Hybrids 1 to 5) according to the *Symphyomyrtus* sections of the component *Eucalyptus* species registered in their pedigrees (Table 2).

## SNP genotyping and filtering

Total genomic DNA was extracted with an optimized Sorbitol/CTAB protocol [37]. DNA samples were sent to ThermoFisher (Santa Clara, CA) for SNP genotyping with the 72K Eucalyptus Axiom Array developed for *Eucalyptus* and *Corymbia* species (https://www.thermofisher.com/order/catalog/product/551134; Grattapaglia D. and Silva-Junior O.B., unpublished). This Axiom Array is a second-generation Eucalyptus SNP platform with 68,055 SNPs specific to the *Eucalyptus* genome, 28,177 of them shared with the previously developed Infinium EUChip60k [31], and 4,147 specific to the genome of its sister genus *Corymbia*, these latter ones not used in this study.

SNPs with more than 10% missing data and with minor allele frequency (MAF) below 5% were removed using PLINK v1.9 [38] using parameters–maf 0.05 –geno 0.1. A total of 48,645 SNPs passed these filtering thresholds. The dataset was further pruned of 21,347 SNPs that were in linkage disequilibrium (LD) with other markers to remove redundant information

**Table 2. List of the 44 *Eucalyptus* hybrids studied, obtained by controlled interspecific crosses of two or more species, classified into five groups according to the sections of *Symphyomyrtus* involved in the cross (Hybrids 1 –Hybrids 5).**

| Hybrid | Sections involved | Individual ID | Parental species crossed |
|---|---|---|---|
| Hybrids 1 | *Latoangulatae x Exsertaria* | *Hyb-1* | *E. urophylla x E. camaldulensis* |
| Hybrids 1 | | Hyb-2 | *E. urophylla x E. camaldulensis* |
| Hybrids 1 | | Hyb-3 | *E. urophylla x E. camaldulensis* |
| Hybrids 1 | | Hyb-4 | *E. urophylla x E. camaldulensis* |
| Hybrids 1 | | Hyb-5 | *E. urophylla x E. camaldulensis* |
| Hybrids 1 | | Hyb-6 | *E. urophylla x E. camaldulensis* |
| Hybrids 1 | | Hyb-7 | *E. urophylla x E. camaldulensis* |
| Hybrids 1 | | Hyb-8 | *E. urophylla x E. camaldulensis* |
| Hybrids 1 | | Hyb-9 | *E. urophylla x E. camaldulensis* |
| Hybrids 1 | | Hyb-10 | *E. urophylla x E. camaldulensis* |
| Hybrids 1 | | Hyb-11 | *E. urophylla x E. camaldulensis* |
| Hybrids 1 | | Hyb-12 | *E. camaldulensis x E. grandis* |
| Hybrids 2 | *Latoangulatae x Maidenaria* | Hyb-13 | *E. dunnii x E. urophylla* |
| Hybrids 2 | | Hyb-14 | *(E. dunnii x E. grandis) x E. dunnii* |
| Hybrids 2 | | Hyb-15 | *(E. grandis x E. saligna) x (E. urophylla x E. globulus)* |
| Hybrids 2 | | Hyb-16 | *((E. grandis x E. urophylla) x (E. grandis x E.globulus)) x ((E. dunnii x E. grandis) x (E. urophylla x E. globulus))* |
| Hybrids 2 | | Hyb-17 | *E. dunnii x E. urophylla* |
| Hybrids 2 | | Hyb-18 | *E. dunnii x E. urophylla* |
| Hybrids 2 | | Hyb-19 | *E. dunnii x E. urophylla* |
| Hybrids 2 | | Hyb-20 | *(E. grandis x E. urophylla) x (E. urophylla x E. globulus)* |
| Hybrids 2 | | Hyb-21 | *E. grandis x E. globulus* |
| Hybrids 2 | | Hyb-22 | *(E. urophylla x E. globulus) x (E. dunnii x E. grandis)* |
| Hybrids 2 | | Hyb-23 | *E. saligna x (E. grandis x E. globulus)* |
| Hybrids 2 | | Hyb-24 | *E. urophylla x (E. grandis x E. globulus)* |
| Hybrids 2 | | Hyb-25 | *(E. dunnii x E. grandis) x E.globulus* |
| Hybrids 2 | | Hyb-26 | *E. saligna x E. globulus* |
| Hybrids 2 | | Hyb-27 | *((E. grandis x E. urophylla) x (E. grandis x E. globulus)) x E. benthamii* |
| Hybrids 2 | | Hyb-28 | *(E. urophylla x E. grandis) x ((E. dunnii x E. grandis) x (E. urophylla x E. globulus))* |
| Hybrids 2 | | Hyb-29 | *(E. urophylla x E. grandis) x E. benthamii* |
| Hybrids 2 | | Hyb-30 | *E. grandis x E. benthamii* |
| Hybrids 2 | | Hyb-31 | *(E. urophylla x E. grandis) x E. globulus* |
| Hybrids 2 | | Hyb-32 | *E. urophylla x E. benthamii* |
| Hybrids 2 | | Hyb-33 | *E. urophylla x E. dunnii* |
| Hybrids 2 | | Hyb-34 | *E. urophylla x E. dunnii* |
| Hybrids 2 | | Hyb-35 | *E. urophylla x E. globulus* |
| Hybrids 2 | | Hyb-36 | *E. urophylla x E. globulus* |
| Hybrids 3 | *Latoangulatae* | Hyb-37 | *E. grandis x E. urophylla* |
| Hybrids 3 | | Hyb-38 | *E. grandis x E. pellita* |
| Hybrids 3 | | Hyb-39 | *E. urophylla x E. grandis* |
| Hybrids 3 | | Hyb-40 | *E. urophylla x E. grandis* |
| Hybrids 3 | | Hyb-41 | *E. urophylla x E. saligna* |
| Hybrids 4 | [a] | Hyb-42 | *E. dunnii x E. globulus* |
| Hybrids 5 | [b] | Hyb-43 | *(E. dunnii x E. grandis) x E. camaldulensis* |
| Hybrids 5 | | Hyb-44 | *E. camaldulensis x (E. urophylla x E. globulus)* |

[a] *Maidenaria x Maidenaria;*

[b] *Exsertaria x Latoangulatae x Maidenaria*

and avoid regions of the genome with a disproportionate influence on the results, that could potentially distort the representation of genome-wide structure [39]. LD pruning was performed using PLINK parameter–indep-pairwise 50 5 0.3. With the retained 27,298 SNPs, the rate of per individual missing data was below 10% for all samples, except for one sample of *E. grandis* from Coffs Harbor. This sample had 64.9% missing genotypes and was removed from further analyses. Ultimately, genetic analyses were performed with a dataset of 27,298 SNPs genotyped in 484 individual trees.

## Statistical and population genetics analyses

Basic population genetics parameters were estimated, such as the average minor allele frequency (MAF), observed ($H_o$) and expected heterozygosity ($H_e$). Analyses were performed in R (R Core Team 2020) using packages adegenet v2.1.3 [40] and hierfstat v 0.5–7 [41]. The data was input into R in FSTAT format after transformation with PGDSpider v2.1.1.5 [42].

fastSTRUCTURE v. 1.0 [43] was run with the 27,298 SNPs to infer population structure for the 484 individual trees. Analyses were performed with the number of clusters K varying from 2 to 30 and option—seed = 100. The input was the binary version (BED) of the PED file from PLINK. The most likely model was selected using the supervised estimators of [44] implemented in the StructureSelector [45] web server (https://lmme.ac.cn/StructureSelector/). Cluster assignment for each of the samples was visualized with barplots in R, using packages pophelper v2.3.0 [46] and gridextra v2.3 [47]. Additional fastSTRUCTURE analyses were carried out separately for individual eucalypt sections and species to assess resolution at within taxa levels for provenances differentiation.

Genomic composition of the hybrids was initially obtained from the unsupervised inference provided by fastSTRUCTURE, and compared with the recorded pedigree information. Specifically, for fastSTRUCTURE annotation, we used the meanQ file, which provided the probabilities of each sample belonging to each of the clusters found. Subsequently, a supervised analysis was carried out using ADMIXTURE, a software for model-based estimation of ancestry in unrelated individuals [48]. For this analysis, samples defined as being from pure species with ~99% probability in the initial fastSTRUCTURE analysis, were used as reference populations to infer the genomic composition of the hybrids' genomes. A simple matching genetic distance among individual trees was also estimated and groups visualized with a principal component analysis (PCA) on the genetic distance matrix, where distances among trees were represented in a cartesian graph with PC1 and PC2. These analyses were performed in R using packages adegenet [40] ape v5.4 [49] and pegas v0.14 [50]. PCA biplots were visualized using ggplot2 v.3.3.2 package [51].

## Results

### SNP diversity across species

After filtering and LD pruning, the final SNP dataset of 27,298 SNPs (S1 File) had a very low percentage of missing data (<3%) for all germplasm sets (species, provenances and hybrids), corroborating the good performance of the multi-species SNP array for population genomics and molecular breeding across eucalypt taxa. The percentage of polymorphic loci per population ranged from 29.7% for *E. deglupta* to over 93% for *E. urophylla* and the hybrids involving crosses between distant sections *Latoangulatae* and *Maidenaria* (Table 3). Overall, there was no significant difference in the proportion of polymorphic SNPs among the different eucalypt sections (ANOVA F-value = 1.14, p-value = 0.37). The average MAF across taxa was similar, within 0.1 and 0.15 for most taxa but *E. urophylla*, *E. grandis*, *E. camaldulensis* and the hybrids had a slightly higher average MAF.

**Table 3. Summary of the proportion of polymorphic SNPs (Minor Allele Frequency; MAF> 0.05) and their average MAF for the 27,298 filtered and LD pruned SNPs, and genetic diversity parameters (observed ($H_o$) e expected ($H_e$) heterozygosity) for each species and hybrid (see Table 2) germplasm source.**

| Species (Provenance) | N | % SNPs MAF>0.05 | Average MAF | Average $H_e$ | Average $H_o$ |
|---|---|---|---|---|---|
| *E. argophloia* | 16 | 42.23% | 0.104 | 0.136 | 0.188 |
| *E. benthamii* | 24 | 42.63% | 0.112 | 0.151 | 0.175 |
| *E. brassiana* | 18 | 55.34% | 0.124 | 0.167 | 0.167 |
| *E. camaldulensis* | 26 | 68.88% | 0.151 | 0.205 | 0.200 |
| *E. deglupta* | 18 | 29.81% | 0.091 | 0.114 | 0.179 |
| *E. dunnii* | 28 | 55.54% | 0.119 | 0.161 | 0.166 |
| *E. globulus* | 26 | 58.34% | 0.128 | 0.172 | 0.176 |
| *E. grandis* (Atherton) | 16 | 68.02% | 0.162 | 0.219 | 0.214 |
| *E. grandis* (Coffs Harbour) | 20 | 79.03% | 0.180 | 0.245 | 0.240 |
| *E. longirostrata* (Coominglah) | 12 | 42.90% | 0.098 | 0.132 | 0.156 |
| *E. longirostrata* (Goodger) | 12 | 41.38% | 0.095 | 0.128 | 0.155 |
| *E. longirostrata* (Starkvale) | 12 | 42.19% | 0.096 | 0.130 | 0.150 |
| *E. nitens* | 20 | 38.94% | 0.083 | 0.336 | 0.158 |
| *E. pellita* (Hopevale) | 16 | 61.56% | 0.132 | 0.180 | 0.182 |
| *E. pellita* (Kuranda) | 15 | 61.28% | 0.137 | 0.186 | 0.168 |
| *E. robusta* (Brisbane) | 16 | 56.76% | 0.125 | 0.170 | 0.183 |
| *E. robusta* (Byfield) | 14 | 51.61% | 0.120 | 0.161 | 0.174 |
| *E. saligna* (Helidon) | 12 | 59.54% | 0.142 | 0.191 | 0.187 |
| *E. saligna* (Kroombit Tops) | 12 | 60.92% | 0.142 | 0.192 | 0.209 |
| *E. saligna* (Richmond Range) | 12 | 60.56% | 0.143 | 0.193 | 0.202 |
| *E. tereticornis* | 7 | 44.83% | 0.120 | 0.385 | 0.198 |
| *E. tereticornis* (Mount Garnet) | 10 | 58.40% | 0.143 | 0.192 | 0.209 |
| *E. tereticornis* (Mareeba) | 12 | 61.09% | 0.146 | 0.196 | 0.209 |
| *E. tereticornis* (Mitchell R.Oaky C) | 12 | 60.10% | 0.144 | 0.193 | 0.209 |
| *E. urophylla* | 42 | 93.67% | 0.219 | 0.297 | 0.272 |
| *E. viminalis* | 12 | 60.16% | 0.122 | 0.166 | 0.178 |
| Hybrids 1 | 12 | 90.29% | 0.225 | 0.304 | 0.307 |
| Hybrids 2 | 24 | 93.94% | 0.229 | 0.309 | 0.318 |
| Hybrids 3 | 5 | 74.70% | 0.205 | 0.272 | 0.316 |
| Hybrids 4 | 1 | 32.43% | 0.163 | 0.188 | 0.334 |
| Hybrids 5 | 2 | 54.22% | 0.186 | 0.231 | 0.333 |

## Population structure analysis

StructureSelector analysis of the fastSTRUCTURE results indicated the most likely model with K = 18 taxonomic clusters (S2 File). This model correctly assigned each of the 16 species to a different cluster (Fig 2; S3 File). In the case of *E. saligna* some individuals were additionally separated according to provenance and the hybrids were assembled in a separate highly admixed cluster (Fig 2). Admixture at the individual level was seen in allegedly pure species trees. Some *E. camaldulensis* individuals were classified as being admixed with *E. tereticornis*, some *E. urophylla* individuals admixed with *E. grandis*, and a few additional admixed individuals were seen that were expected pure (Fig 2). At the higher taxonomic level of sections within subgenus *Symphyomyrtus*, models with smaller numbers of clusters easily separated eucalypt sections. For example, at K = 2, section *Maidenaria* detached from the rest. With K = 3, *Latoangulatae* and *Maidenaria* split, with occasional admixture seen in individuals of some species. With K = 4, species of *Exsertaria* separated from the other sections. Surprisingly,

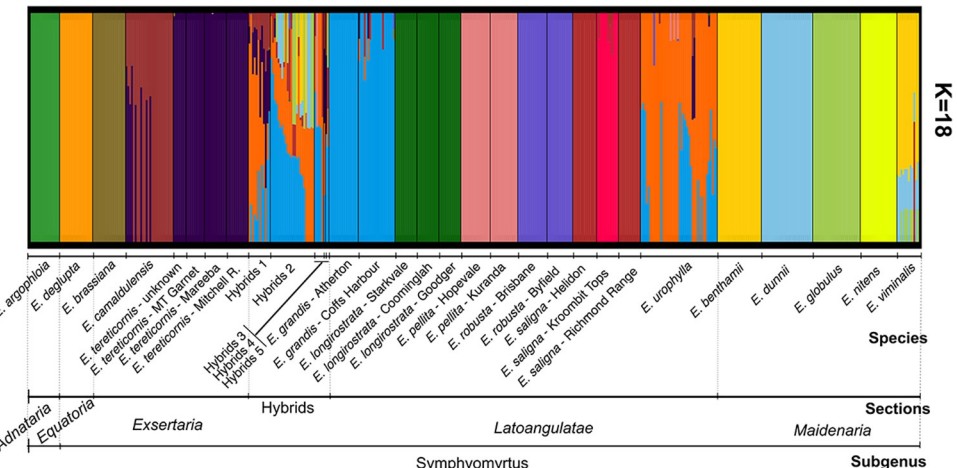

**Fig 2. Population structure analysis of the 440 trees of 16 *Eucalyptus* species and 44 hybrids classified according to the section of their component species involved (Hyb 1 to 5).**

however, *E. longirostrata* that belongs to section *Exsertaria*, was clustered together with *E. deglupta* and *E. argophloia* that belong to two other different sections (S4 File).

The SNPs dataset could not differentiate most provenances within species when all individuals were analyzed together, except for *E. saligna* from Kroombit Tops (Fig 2). This provenance was assigned to a separate group from Helidon and Richmond Range provenances, which in turn were clustered together. All the provenances for the other species (*E. tereticornis*, *E. grandis*, *E. longirostrata*, *E. pellita*, *E. pilularis* and *E. robusta*) could not be discriminated even at higher K's (S5 File). Only when species were analyzed individually, fastSTRUCTURE modeling resolved some of the provenances. This was the case of the two *E. grandis* provenances from Atherton and Coffs Harbor and *E. robusta* from Brisbane and Byfield. Somewhat separate clustering was also seen for *E. longirostrata* from Starkvale, and *E. tereticornis* from Mount Garnet, although some individuals either displayed admixture or were not clustered accordingly (Fig 3). Lastly, provenances of some species clearly could not be distinguished. This occurred with *E. pellita*, *E. longirostrata* from Coominglah and Goodger and with *E. tereticornis* from Mitchell Road (Oaky Creek) and Mareeba.

## Determination of ancestral species composition of hybrids

The ancestral genomic composition of hybrids estimated with both fastSTRUCTURE and ADMIXTURE were compared to their respective pedigree expected composition (Fig 4; S6 File). The supervised analysis carried out using ADMIXTURE resulted, in general, in similar genomic composition as those obtained with fastSTRUCTURE, although some differences were seen for example in Hybrids 1, where the fastSTRUCTURE model indicated the unexpected presence of *E. tereticornis* genome. Overall, there were only nine out of the 41 hybrids for which the SNP-based composition closely matched the pedigree expected one. This happened for hybrids Hyb-31, Hyb-32, Hyb-33, Hyb-34, Hy-35, Hyb-36, Hyb-38, Hyb-39, Hyb-40 e Hyb-41, almost all of them simple $F_1$ hybrids. For all other hybrids, small to large deviations were observed.

For a considerable number of hybrids, additional unanticipated species from those recorded in the pedigree, were observed in their composition (Fig 4). For example, hybrids Hyb-1 through Hyb-11 in the Hybrids 1 group were expected to be $F_1$'s of *E. urophylla* and *E.*

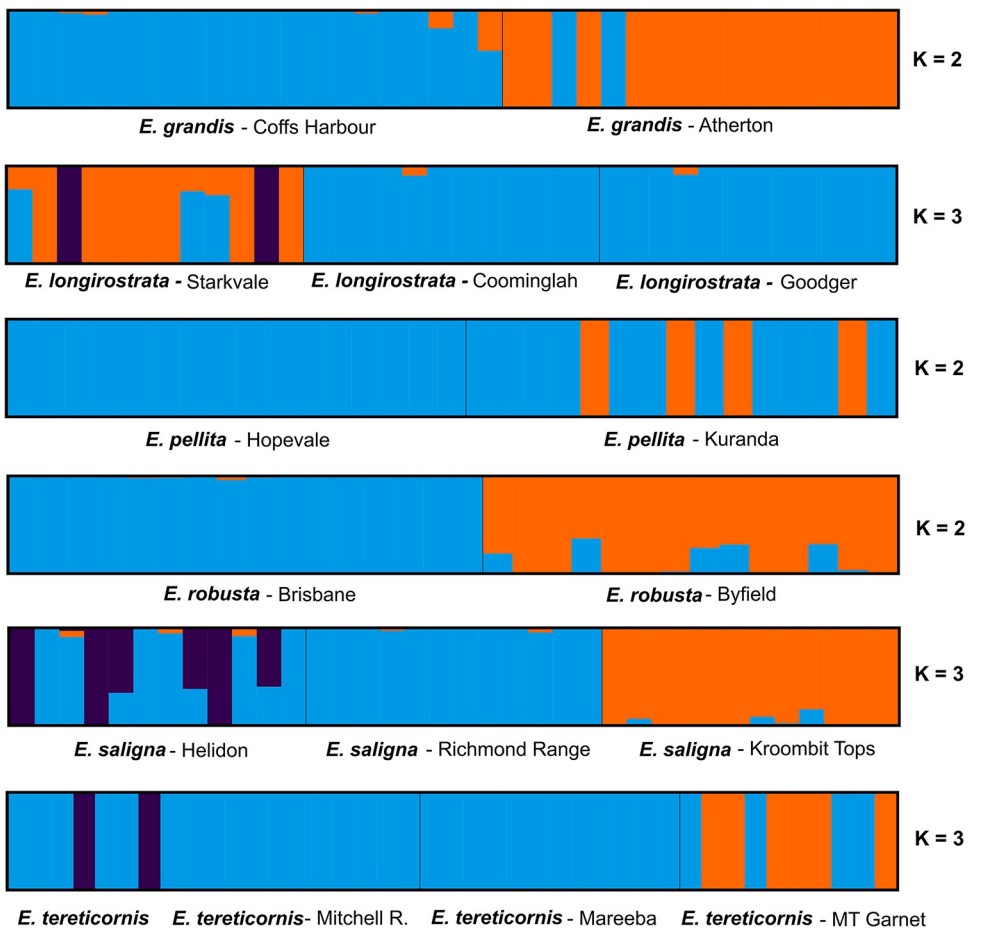

**Fig 3. Population structure analysis of each *Eucalyptus* species separately for which more than one provenance was studied.**

*camaldulensis*. However, eight of them showed variable amounts of *E. grandis* genome in the ADMIXTURE analysis while the fastSTRUCTURE model suggested the presence of *E. tereticornis* genome more frequently than that of *E. camaldulensis*. The unexpected presence of *E. grandis* genome was again seen in several other hybrids in the Hybrids 2 group (ex. Hyb-17, Hyb-18, Hyb-19, Hyb-26). Furthermore, in this group of hybrids none or a considerably less than expected proportion of the genome was detected coming from the recorded species of *Maidenaria* involved in the crosses, namely *E. dunnii* and *E. globulus*. *E. dunnii* genome was not detected in six of the 14 hybrids and *E. globulus* in seven of 12 where it should have been observed (Fig 4). For example, in hybrids Hyb-13, Hyb-17, Hyb-18, Hyb-19, Hyb-21 and Hyb-26 expected to be $F_1$ hybrids of *Latoangulatae* species (*E. urophylla*, *E. grandis* or *E. saligna)* with *Maidenaria* species *(E. dunnii* or *E. globulus*), the SNP data showed little or no sign of the two temperate species genomes and an unexpected or larger than expected proportions of the genome of *E. grandis*. Finally, there were cases where the presumed genomic composition was completely different from the SNP-estimated one. For example, hybrid Hyb-42 was expected to be a *E. dunnii* x *E. globulus* hybrid, when in fact it involved mainly species of *Latoangulatae* with *E. camaldulensis*, suggesting mislabeling.

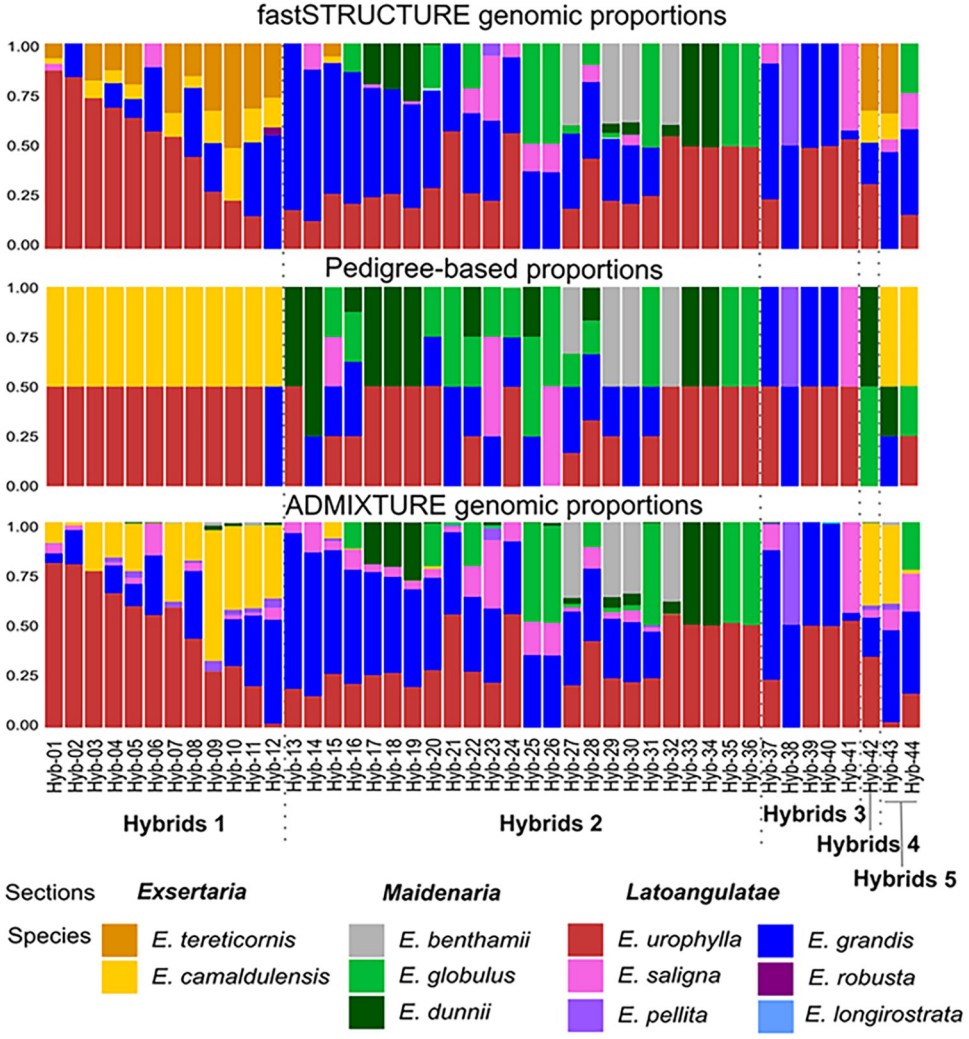

**Fig 4. Analysis of the ancestral species' genomic composition of the 44 interspecific hybrids studied.** The genomic proportions estimated by unsupervised inference with fastSTRUCTURE (top panel) and by a supervised model with species data as reference with ADMIXTURE (bottom panel), were compared with the expected composition from pedigree information (middle panel). The species were categorized into sections according to Brooker's (2000) classification.

## Genetic distances among species, provenances and hybrids

Overall, the PCA plot based on the genetic distance matrix positioned the different species and sections as expected, clustering phylogenetically closer species of the same section (Fig 5). A clear exception, however, was seen for *E. longirostrata*, taxonomically classified in section *Latoangulatae*. The PCA placed it away from *Latoangulatae* and closer to *E. argophloia* and *E. deglupta*. These two species belong to two different sections but they clustered together, considerably separated from all other species. In most cases, the PCA analysis had no resolution to discriminate provenances within species. In line with the fastSTRUCTURE results, exceptions were the Kroombit Tops provenance of *E. saligna*, and the two provenances each of *E. robusta* and *E. grandis* that were separated in the PCA.

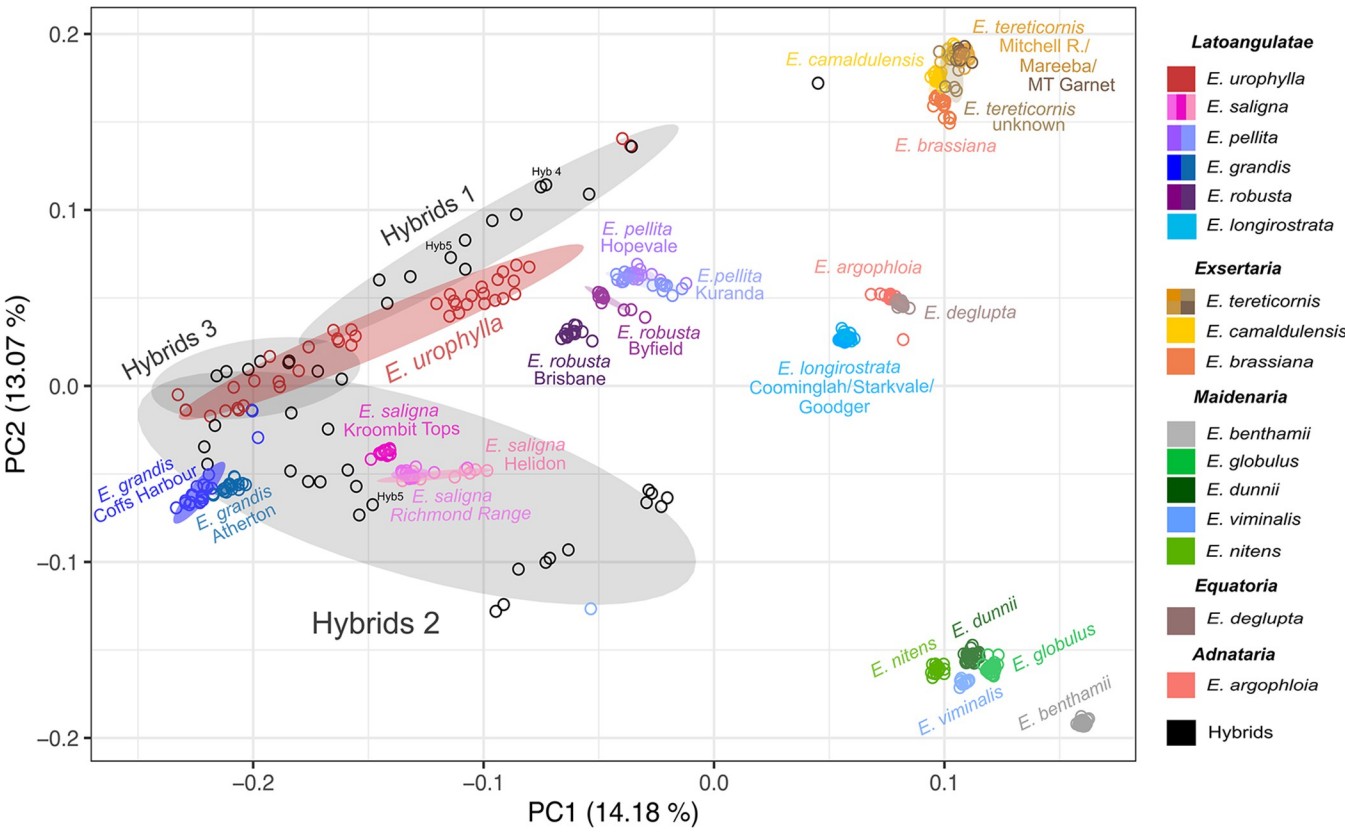

**Fig 5. PCA scatter plot of the 484 *Eucalyptus* individuals in the first two principal components.** Samples colored by species, provenances and hybrids were grouped in their respective classes according to the taxonomic sections involved in the cross. The ellipses depict the 95% confidence interval for the distribution of each species or hybrid group.

## Discussion

### Genome-wide eucalypt species SNPs diversity

Consistent with the initial validation data provided alongside the EuCHIP60k development [31], our results corroborate that the current SNPs arrays platforms offer effective power to carry out genetic diversity analysis of the main eucalypt species planted worldwide. Within species, the proportion of polymorphic SNPs showed some variation, although for the vast majority, over 40% of the SNPs were informative and the average MAF was generally above 0.13, despite the relatively limited sample sizes analyzed (Table 3). Higher proportions of polymorphic SNPs, above 68% up to 93%, and higher average MAF were observed for *E. grandis*, *E. camaldulensis* and *E. urophylla*. These results may be explained in part from the somewhat larger sample sizes analyzed. In the case of *E. urophylla* the alleged mixture of provenances might have contributed to the higher diversity. A second explanation for the higher SNP diversity in these three species involves potential admixture due to unintended interspecific hybridization. These three species are widely used to generate interspecific hybrids in Brazil and the structure analysis results indicated admixture in the *E. urophylla* and *E. camaldulensis* trees (see below).

A third possible explanation for the higher SNP diversity observed in *E. grandis*, *E. camaldulensis* and *E. urophylla* is some ascertainment bias derived from the discovery panels used in the initial SNP discovery for the development of the EuCHIP60K. Although SNP discovery was carried out on sequence data for 240 trees of 12 species, a proportionally larger amount of

sequence data was obtained for these three species when compared to the others [31]. Large proportions of informative SNPs (58–60%) were also seen for species of *Maidenaria*, consistent with the fact *that E. globulus* was also an important target of sequence production during SNP discovery. Larger proportions of polymorphic SNPs and higher average MAF were also observed in the different hybrids. This was evidently expected, given the transmission to the hybrid of alternative SNP alleles fixed in each parental species.

Except for *E. urophylla*, *E. grandis*, *E. camaldulensis* and the hybrids, the results suggest that the rate of SNP polymorphism might depend more on the level of genetic diversity captured in the specific sample of individuals than on the particular species analyzed. This in turn indicates that the SNP set used delivers largely equivalent numbers of polymorphic SNPs between any pairwise taxa within the main sections of subgenus *Symphomyrtus*. This indicates good potential for the selection of ancestry informative SNPs sets [52] that appear in substantially different frequencies between species, provenances or populations in this phylogenetic group. The expansion of the number of species and provenances and the specific selection of ancestry informative SNPs at the species and provenances levels would constitute an obvious follow-up of this study.

## SNPs recover the expected species structure but admixture is present

Genome-wide SNP data provided the necessary resolution to check and validate the phylogenetic classification of germplasm sources of the eucalypt species sampled in this study. The most likely model for the SNP dataset found k = 18 clusters, allowing clearcut discrimination of the five sections and the 16 species sampled of subgenus *Symphyomyrtus*, while reliably indicating the admixed composition of hybrids (Fig 2). This result substantiates what a number of previous phylogenetic studies have shown using different types of DNA marker data such as ribosomal ITS, chloroplast DNA, microsatellites and DArT (reviewed in [2]), and more recent studies that further expanded the sampling of taxa and individuals within taxa [8]. The evolutionary history 'written' in the genome of these *Symphyomyrtus* species is generally consistent with their current phylogenetic organization within this subgenus.

Differently from several previous reports that examined germplasm sampled exclusively in their center of origin, our study included material conserved in exotic conditions from variable sources (Table 1). In general, for the species' germplasm that came directly from original sources in Australia, the genetic structure splits were clearcut. For species that included material from unknown origins or collected in germplasm banks established in Brazil, occasional admixture was seen. The *E. camaldulensis* trees showed admixture with *E. tereticornis*, *E. viminalis* individuals showed admixture with *E. dunnii*, and *E. urophylla* sampled from multiple provenances established in Brazil displayed significant admixture with *E. grandis* (Fig 2). For some of these germplasm sources our data indicate that accidental hybridization might have taken place once the germplasm was introduced in Brazil. In the exotic habitat under different ecoclimatic conditions, reproductive barriers between eucalypt species such as geographic distance and flowering phenology that maintain species apart in their natural range, are relaxed or even broken, facilitating hybridization [53]. The paradigmatic example is the famous eucalypt hybrid swarm of the Rio Claro Arboretum established upon the introduction of *Eucalyptus* species in Brazil in 1904 [53, 54]. Several species were planted side by side, and seeds collected from that germplasm generated very heterogenous plantation forests, where some hybrids of unknown origin and outstanding performance were selected and are still planted or used in breeding programs today [17, 18]. The results of our study point to the development of ancestry informative SNPs that should allow reconstructing and understanding the recombination history of these hybrids.

The *E. viminalis* germplasm sample also showed evidences of admixture with *E. dunni* and *E. globulus* at k = 18. This sample of trees was from an advanced generation germplasm source established in Brazil but with unknown origin in Australia. Hybridization between these temperate species of section *Maidenaria* once introduced in Brazil cannot be ruled out, although less likely than for the previously mentioned species of *Exsertaria* and *Latoangulatae*, since *Maidenaria* species flower and produce seed less conspicuously in the tropics [53]. When a model with k = 20 was tested, *E. viminalis* individuals clearly split (S5 File) with no evidence of hybrid constitution. This result highlights the long-standing challenge with admixture modeling, whereby the most likely selection of K clusters is a difficult problem to automate in a way that is effectively robust [39].

The graphical projection of the different species and hybrids in the PCA was generally consistent with the phylogenetic expectations (Fig 5). Complementing the structure analysis, the PCA provided additional information regarding the genetic distance among the different taxa. *E. deglupta* and *E. argophloia* were placed at a considerable distance from the main section of *Symphyomyrtus*. The fact that they clustered together was however unexpected, since they are classified in distinct sections. These two species are currently part of *Symphyomyrtus* [55] and while no contention exists regarding the classification of *E. argophloia*, *E. deglupta* has originally been classified in subgenus *Minutifructis* [4]. The three main sections of interest in the subgenus were clearly separated and contained the expected species, exception made for *E. longirostrata* that clustered away from its section *Latoangulatae* and distant from *Exsertaria* as well.

Samples of *E. longirostrata* have been examined in the most extended molecular phylogenetic study of terminal taxa of sections *Maidenaria*, *Exsertaria* and *Latoangulatae* to date [8]. That study produced a phylogeny that largely matched the morphological treatment of sections, although sections *Exsertaria* and *Latoangulatae* were shown to be polyphyletic. Several inconsistencies between the morphological classification and the molecular phylogeny were described, and a number of taxa in *Latoangulatae* were deemed polyphyletic at the species level. A polyphyletic group is one that shows mixed evolutionary origin, descended from more than one ancestor, with taxa sharing homoplasies, typically explained as a result of convergent evolution, complicating the correct taxonomical classification [56]. *E. longirostrata* was itself deemed polyphyletic, classified within series *Lepidotae-Fimbriatae* and clustered into *Latoangulatae* IV, a clade considerably distant from *Latoangulatae* II where *E. grandis*, *E. pellita*, *E. robusta* and the section type species *E. saligna* belong. Furthermore, those authors suggest that all *Latoangulatae* species other than those in *Latoangulatae* II would be better placed in other taxonomic sections to reflect the phylogeny revealed in their study. The most recent classification of the eucalypts [14, 55] however, classified *E. longirostrata* into a different section, *Pumilio*. In our study, the sharp split of *E. longirostrata* from *Latoangulatae* and *Exsertaria* (Fig 5), provides further molecular evidence for this most recent taxonomic classification placing the species in a separate section.

## Provenance discrimination is strongly dependent on geographical distance

With the exception of one provenance of *E. saligna*, all other Eucalyptus provenances could not be discriminated when all 484 samples were analyzed together (Fig 2). When species were analyzed separately, provenances could be discriminated for some species but not for others (Figs 3, 5). Looking at the geographical position of the sampled provenances (Fig 1), a pattern emerged suggesting that SNP-based discrimination was strongly dependent of geographical distance. The two provenances of *E. grandis* (Atherton and Coffs Harbor), separated in the structure and PCA analyses, are located at more than 2,000 km apart. The same happened with

provenances Byfield and Brisbane of *E. robusta* at ~700 km from each other, and *E. saligna* Kroombit Tops provenance located at >700 km from the other two *E. saligna* provenances. All other provenances that were loosely or otherwise not discriminated are located at less than 200–300 km apart. These results indicate an isolation-by-distance (IBD) model of population structure for the provenances sampled for these species. The genetic similarity between populations will decrease exponentially as the geographic distance between them increases, because of the limiting effect of geographic distance on rates of gene flow [57].

A number of studies in *Eucalyptus* have looked at the prevalence of genetic structure between populations located at various geographic distances. These studies have generally shown that an IBD model fits well the observed data, with genetic distances between provenances strongly positively correlated with geographic distances [24, 58, 59]. A recent landscape study based on very dense DNA data obtained by whole genome sequencing in *E. albens and E. sideroxylon*, also found strong support for IBD in both species [60]. Taken together, ours and others' results indicate that clearcut distinction of *Eucalyptus* germplasm sources in what regards provenance variation, might not be straightforward even with a dense panel of SNPs, unless provenances are geographically distant or provenance-informative SNP markers are specifically identified and used. As a result, what breeders may call as different provenances could in effect be members of the same continuous population despite several kilometers of physical distance, if gene flow is ubiquitous. It must be mentioned, however, that our study suffered from limited and somewhat uneven sampling of provenances that might have contributed to a greater difficulty in distinguishing some of them. It has been shown that subpopulations with reduced sampling tend to be merged together in genetic structure analyses, and uneven sampling may lead to downward-biased estimates of the true number of subpopulations [44]. Larger sample sizes for the provenances studied should allow better estimation of allele frequencies and possibly selection of ancestry informative, provenance-specific SNPs for greater discrimination power.

## Genomic composition of hybrids indicates directional selection toward tropical genomes

Our genome-wide data showed that the majority of the hybrids studied (35 out of 44) displayed genomic composition deviating from the expected one based on pedigree information (Fig 4). This result is important in view of the long standing and widespread adoption of deliberate breeding strategies toward the selection of elite hybrid clones with specific anticipated genomic composition, especially in tropical countries (reviewed in [12]). This in turn highlights one more important application of using dense, high-quality array-based SNP data in support of breeding programs. SNP data not only provide precise germplasm identity verification, but more importantly allow the breeder to objectively recognize the actual ancestral origin of superior hybrids in order to discard unwanted hybrid combinations or to more realistically guide the breeding program toward the development of the desired genetic material.

For the sample of hybrids studied in this work, the lack of adherence between the expected genomic composition and the actual one suggests at least two hypotheses. Notwithstanding the possibility of mislabeling errors during controlled crosses, as likely the case for hybrids Hyb-13, Hyb-14 and Hyb-42, the second and most probable hypothesis is pervasive genetic admixture of the parents involved in the original interspecific cross. Given the frequently unknown introduction history, followed by local intermating in Brazil in the last 120 years, as discussed previously, there is a considerable possibility that the presumed parents were themselves misclassified. Moreover, because hybrids tend to be produced by crossing good performing parents in the breeding program, it is quite possible that actually some of the parents

used were themselves hybrids, distorting the expected composition of the resulting hybrid off-spring. Species within the same sections of *Symphyomyrtus* that display overlapping morphological features and easily hybridize would be more prone to such occurrences. Clearcut examples were six supposedly $F_1$ hybrids that in principle did not involve *E. grandis*, but where the SNP data revealed its presence (Hyb-13, Hyb-17, Hyb-18, Hyb-19, Hyb-26, Hyb-42). Likewise, several $F_1$ hybrids of *E. urophylla* with *E. camaldulensis* (Hybrids 1 group) showed variable amounts of *E. grandis* genome in their composition, and the presence of *E. tereticornis* genome more frequently than that of *E. camaldulensis* (Fig 4). Admixture of *E. grandis* genome into the *E. urophylla* parents and difficulties in morphologically discriminating *E. camaldulensis* germplasm from *E. tereticornis* could readily explain these results.

Besides the presence of *E. grandis* as an unexpected species in the genomically realized pedigree, the observation of larger than expected proportions of *E. grandis* genome was also seen for all hybrids where this species was involved. Fourteen hybrids derived from advanced generation recombinant intercrosses involved one or both hybrid parents with three or more species represented, *E. grandis* being one of them (ex. Hyb-14, Hyb-15, Hyb-16, Hyb-20, Hyb-22 through Hyb-25, Hyb-27 through Hyb-29, Hyb-32, Hyb-43 and Hyb-44) (Table 2). The pedigree-expected proportions were estimated based on the final presumed participation of each single species in the pedigree, assuming balanced Mendelian inheritance and recombination rates in the previous hybrid generations with no selection. For all these 20 hybrids, the SNP data showed, however, a consistently higher proportion of *E. grandis* genome in the hybrid composition. Aside from unintended admixture in the original parents, the ubiquitous unexpected presence or higher than anticipated proportion of *E. grandis* genome in the vast majority of hybrids, strongly suggests genome-wide directional selection for this species' genome throughout the breeding history of these complex hybrid clones. This should not be surprising given that volume growth is the main breeding target, and that *E. grandis* is well known for its fast growth [53]. Our data therefore not only corroborates the pivotal role of *E. grandis* in hybrid breeding, but also shows that its actual participation is considerably larger than expected and frequently unintended. Moreover, our data also demonstrate that in hybrids between species of *Latoangulatae* and *Exsertaria* with species of *Maidenaria* (Hybrids 2 group), the actual participation of the latter, such as *E. globulus. E. dunnii* and *E. benthamii* in the final hybrid's genome composition is less than expected, consistent with strong selection against the less adapted temperate genomes in tropical environments.

## Concluding remarks

In conclusion, we have shown that the current *Eucalyptus* multi-species SNP array platform, provides a valuable tool to look at within taxa variation in *Symphyomyrtus*, to investigate population structure and track the genomic ancestry of individual clones. As the current "gold standard" in the high-throughput SNP genotyping industry, SNP arrays provide full data portability across studies carried out at different times. This represents a crucial advantage for the construction of legacy SNP databases for multiple *Eucalyptus* species and populations when compared to reduced representation genotyping by sequencing methods. SNP array data portability across studies allows effortless data consolidation across time for comparative studies and meta-analyses, that should be valuable for resolving taxonomic issues that still persist in the eucalypts. We are aware, however, that for eucalypt species phylogenetically distant from subgenus *Symphyomyrtus*, the current SNP array will not provide equivalent numbers of informative SNPs due to a higher genomic divergence [31].

We have also shown that while species classification is well resolved at the genome-wide level, provenance discrimination is not always so. It depends essentially on geographical

distance, consistent with an isolation by distance model, and likely to be impacted by sample size. Further studies with larger samples sizes and the identification of provenance specific SNPs are warranted. Finally, our results are novel in that they objectively show, based on SNP data, that unplanned genetic admixture should not be a surprise in exotic germplasm sources not only in Brazil but likely in other countries, especially among phylogenetically closer species that easily hybridize in exotic environments. Moreover, the genomic ancestral composition of control-crossed hybrids in Brazil indicated that strong selection takes place in favor of tropical genomes and more specifically that of *E. grandis*. SNP-based auditing of hybrids' genomic composition could be introduced as a standard practice in hybrid breeding programs to more truthfully guide the program toward the development of the desired genetic material.

## Supporting information

**S1 File. SNP genotype data.** Complete dataset for the filtered 27,298 SNPs obtained with the 72k Eucalyptus Axiom Array for the 484 individuals studied.
(CSV)

**S2 File. Supervised estimators of k clusters.** Results of the four supervised estimators of Puechmaille (2016) to detect the number of clusters implemented in the web server Structure-Selector (Li and Liu 2018) indicating that the germplasm set is most likely structured in 18 clusters after modelling with a variable number of k from 2 to 30 using FastStructure.
(DOCX)

**S3 File. Output fastSTRUCTURE at k = 18.** Output of the meanQ values of the fastSTRUC-TURE analysis of the 484 individuals studied with K = 18.
(XLSX)

**S4 File. Structure analysis plot at k = 2 to 4.** Population structure analyses of the *Eucalyptus* species and hybrids clustered with variable numbers of clusters (*K*) from 2 to 4 separating the *Eucalyptus* sections (*Maidenaria*, *Latoangulatae* e *Exsertaria*), while displaying admixture in species of section *Latoangulatae*.
(DOCX)

**S5 File. Structure analysis plot at k = 20 to 30.** Population structure analyses of the *Eucalyptus* species and hybrids clustered with variable numbers of clusters (K) from 20 to 30, beyond the most likely model with K = 18.
(DOCX)

**S6 File. Output fastSTRUCTURE & ADMIXTURE of hybrids' composition.** Output of the meanQ values of the unsupervised fastSTRUCTURE analysis (sheet A) and supervised ADMIXTURE analysis (sheet B) of the ancestral genomic composition of the 44 hybrids.
(XLSX)

## Acknowledgments

We would like to thank the IPEF cooperative tree breeding program (PCMF) affiliated companies, highlighting CMPC, Klabin, Suzano and Vallourec for providing materials for the study. Special thanks to the Experimental Stations of Forestry Sciences at ESALQ/USP for maintaining a large genetic collection of eucalypts in partnership with IPEF (current agreement: 1013868) and to prof. Alexandre S. Coelho for his assistance with computational facilities.

## Author Contributions

**Conceptualization:** Paulo Henrique Muller da Silva, Evandro Novaes, Dario Grattapaglia.

**Data curation:** Danyllo Amaral de Oliveira, Paulo Henrique Muller da Silva, Evandro Novaes, Dario Grattapaglia.

**Formal analysis:** Danyllo Amaral de Oliveira, Paulo Henrique Muller da Silva, Evandro Novaes.

**Funding acquisition:** Paulo Henrique Muller da Silva, Dario Grattapaglia.

**Investigation:** Danyllo Amaral de Oliveira, Evandro Novaes.

**Methodology:** Paulo Henrique Muller da Silva, Evandro Novaes.

**Project administration:** Paulo Henrique Muller da Silva, Evandro Novaes.

**Resources:** Paulo Henrique Muller da Silva, Evandro Novaes, Dario Grattapaglia.

**Software:** Evandro Novaes.

**Supervision:** Evandro Novaes.

**Validation:** Dario Grattapaglia.

**Writing – original draft:** Danyllo Amaral de Oliveira, Dario Grattapaglia.

**Writing – review & editing:** Danyllo Amaral de Oliveira, Evandro Novaes, Dario Grattapaglia.

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
