## [Decision Letter · Decision Letter 0]

14 Jun 2023

PONE-D-23-13755Genome-wide analysis highlights genetic admixture in exotic germplasm resources of Eucalyptus and unexpected ancestral genomic compositions of interspecific hybridsPLOS ONE

Dear Dr. Grattapaglia,

Thank you for submitting your manuscript to PLOS ONE. After careful consideration, we feel that it has merit but does not fully meet PLOS ONE’s publication criteria as it currently stands. Therefore, we invite you to submit a revised version of the manuscript that addresses the points raised during the review process.

We look forward to receiving your revised manuscript.

Kind regards,

Chunxian Chen, Ph.D.

Academic Editor

PLOS ONE

2. We note that Figure 1 in your submission contain [map/satellite] images which may be copyrighted. All PLOS content is published under the Creative Commons Attribution License (CC BY 4.0), which means that the manuscript, images, and Supporting Information files will be freely available online, and any third party is permitted to access, download, copy, distribute, and use these materials in any way, even commercially, with proper attribution. For these reasons, we cannot publish previously copyrighted maps or satellite images created using proprietary data, such as Google software (Google Maps, Street View, and Earth). For more information, see our copyright guidelines: http://journals.plos.org/plosone/s/licenses-and-copyright.

Additional Editor Comments:

This report was to use SNPs to assess the phylogenetic relationship and population structure of a collection of 440 Eucalyptus trees in Brazil, belonging to 16 species and 44 interspecific hybrids. The assessment revealed genetic admixture in exotic germplasm and genomic composition in interspecific hybrids. The paper is well organized, and the SNPs and phylogenetic results are potentially useful for Eucalyptus breeding. Examples of some language, grammar or sentence structure issues are given below, and search and corrections in the entire text would be needed.

Abstract

Line 39, "material" may need a plural form.

Line 44, "composition" may need a plural form.

Introduction

The first sentence may be revised to "Eucalyptus is a genus including highly diverse species ruling the forest across a large part of Australia and neighboring islands."

"in what regards" may be changed to "to".

"taxonomic classification" may be changed to "classification" or "taxonomy".

Results

Table 3, the column "Species (Provenance" may be better viewed with "align left", other than "align center".

Line 293, May "composition" be a plural form? This word is used inconsistently in the paper, in terms of plural or singular form, such as in title, line 294, 297, 301, 306, etc. Similar issue exists with the word "distance" in the paper.

Reviewers' comments:

Reviewer's Responses to Questions

**Comments to the Author**

1. Is the manuscript technically sound, and do the data support the conclusions?

Reviewer #1: Yes

2. Has the statistical analysis been performed appropriately and rigorously? 

Reviewer #1: N/A

3. Have the authors made all data underlying the findings in their manuscript fully available?

Reviewer #1: Yes

4. Is the manuscript presented in an intelligible fashion and written in standard English?

Reviewer #1: Yes

5. Review Comments to the Author

Reviewer #1: In this study, 440 trees including 16 different Eucalyptus species and 44 interspecific hybrids conserved in Brazil were applied for distinguishing the admixture of species and verifying their origins or ancestors by using genome-wide SNP. 27,298 SNPs were identified in 484 individual trees. Population structure and ancestral species composition of hybrids analysis, and genetic distances among species, provenances and hybrids were performed in this study. The results provided precise germplasm identity and the actual ancestral origin of superior hybrids verification, which will be valuable for further breeding in Eucalyptus. Hence, I recommend that the paper was suitable for publishing in the journal. However, there are still several minor revisions.

In material and methods, please provide the details of location of Eucalyptus in Brazil if possible. Maybe it can be added in Table1.

In result and discussion, in most of hybrids, higher proportion of E. grandis genome in the hybrid composition was identified. The authors thought the fast growth was one of reason. As we known, why did E. urophylla known as one of fast growth species not happen in the hybrids?

A large number of sentences were elusive and the quality of the English writing must be improved (example of sentence: L94-97 "However, issues still remain for example for taxa in section Latoangulatae, that still distort the phylogeny due to their intermediate nature when compared to more densely clustered taxa in other sections [8]", and so on.)

6. PLOS authors have the option to publish the peer review history of their article (what does this mean?). If published, this will include your full peer review and any attached files.

Reviewer #1: No

---

## [Author Response · Author response to Decision Letter 0]

30 Jun 2023

Response letter for Revised manuscript PONE-D-23-13755 Oliveira et al.

Genome-wide analysis highlights genetic admixture in exotic germplasm resources of Eucalyptus and unexpected ancestral genomic compositions of interspecific hybrids

Responses to the Editor’s comments

Editor Comments (in bold) and responses (R:) follow thereafter.

R: We have followed closely the PLOS ONE style requirements

2. We note that Figure 1 in your submission contain [map/satellite] images which may be copyrighted. All PLOS content is published under the Creative Commons Attribution License (CC BY 4.0), which means that the manuscript, images, and Supporting Information files will be freely available online, and any third party is permitted to access, download, copy, distribute, and use these materials in any way, even commercially, with proper attribution. For these reasons, we cannot publish previously copyrighted maps or satellite images created using proprietary data, such as Google software (Google Maps, Street View, and Earth). For more information, see our copyright guidelines: http://journals.plos.org/plosone/s/licenses-and-copyright. We require you to either (1) present written permission from the copyright holder to publish these figures specifically under the CC BY 4.0 license, or (2) remove the figures from your submission. In the figure caption of the copyrighted figure, please include the following text: “Reprinted from [ref] under a CC BY license, with permission from [name of publisher], original copyright [original copyright year].”

R: The base map used in the figure was downloaded from "World boundaries" of the World Bank which is free for use under CC_by_4.0 license. The link where the map was obtained is the following https://datacatalog.worldbank.org/search/dataset/0038272/World-Bank-Official-Boundaries. We have improved the text in Material and Methods and figure legend including the details on the source and CC by 4.0 permission of the base map used.

R: We have used EndNote 20 to prepare the references using the PlosONE style and further checked the completeness of the references.

Abstract

Line 39, "material" may need a plural form.

R: Correction made

Line 44, "composition" may need a plural form.

R: Correction made

Introduction

The first sentence may be revised to "Eucalyptus is a genus including highly diverse species ruling the forest across a large part of Australia and neighboring islands."

R: I party agreed with the suggestion. Actually, it is correct that the genus is diverse as it encompasses a very large number of species (more than 900). Furthermore, its ecological relevance goes way beyond forests. As many species eucalypts are not trees but shrubs, the genus is a keystone component across the entire forest landscape. So, the sentence now reads: “Eucalyptus is a highly diverse genus of tree species from Australia and neighboring islands ruling the forest landscape across large part of Australia and neighboring islands.

"in what regards" may be changed to "to".

R: Change made

"taxonomic classification" may be changed to "classification" or "taxonomy".

R: Change made

Results

Table 3, the column "Species (Provenance" may be better viewed with "align left", other than "align center".

R: Change made

Line 293, May "composition" be a plural form? This word is used inconsistently in the paper, in terms of plural or singular form, such as in title, line 294, 297, 301, 306, etc. Similar issue exists with the word "distance" in the paper.

R: Thanks for the observation. The noun “composition” can be plural form. See at: https://www.collinsdictionary.com/dictionary/english/composition#:~:text=(k%C9%92mp%C9%99z%C9%AA%CA%83,Word%20forms%3A%20plural%20compositions)

However, specifically in the text, its use is as an uncountable noun and it should actually be singular, unless a specific syntax agreement to plural was necessary. In just one case we used in plural, on line 118 “… or aim at deliberately exploiting provenance and species complementarity by building specific genomic compositions by interspecific hybridization”. In this case the plural form would be advised as I am referring to different hybrids that will correspond to different compositions.

Responses to Reviewer 1 comments

In material and methods, please provide the details of location of Eucalyptus in Brazil if possible. Maybe it can be added in Table1.

R: Actually, eucalypts are planted throughout the entire country in Brazil. We therefore believe that a map of the locations where eucalypts are planted in Brazil would not be relevant to this particular study, as its focus was on the genetic characterization of germplasm, irrespective where it is planted. 

Regarding the hybrids studied in this work (Table 1) they are all located in a single location at the experimental station of the Institute for Forestry Research (IPEF) as specified in Material & Methods. We added the geographic coordinates or that experimental station. For the germplasm sources of the pure species, the locations are where the germplasm collection sources are actually planted. These locations are not a relevant information as no phenotypic data is used in the study that could be impacted by the specific environment where they are located. The relevant information for the objective of the study is the original location of the seed sources (provenances) and these are shown in detail in Figure 1. 

In result and discussion, in most of hybrids, higher proportion of E. grandis genome in the hybrid composition was identified. The authors thought the fast growth was one of reason. As we known, why did E. urophylla known as one of fast growth species not happen in the hybrids?

R: Actually, almost all hybrids also had a significant proportion of E. urophylla genome as can be clearly seen in Figure 4 (red color). Actually, the proportion of E. urophylla was higher than the 50% expected in most of the hybrids in the first group (Hyb01- through Hyb-11) but lower than expected in hybrids Hyb-17 through Hyb-20. See Figure 4 please.

A large number of sentences were elusive and the quality of the English writing must be improved (example of sentence: L94-97 "However, issues still remain for example for taxa in section Latoangulatae, that still distort the phylogeny due to their intermediate nature when compared to more densely clustered taxa in other sections [8]", and so on.)

R: We improved this sentence and went through the entire manuscript checking for potential unclear sentences and improved when deemed necessary. Furthermore, we had a couple of native speakers check our manuscript for such occurrences and make suggestions for improvements. We have also rearranged some of the texts in the introduction for a more fluid reading and better connection among the paragraphs.

 Thank you for your consideration. We look forward to hearing from you.

Sincerely, 

Dario Grattapaglia

Plant Genetics Laboratory, Embrapa - Recursos Genéticos e Biotecnologia

e-mail: dario.grattapaglia@embrapa.br

---

## [Editor Report · Decision Letter 1]

21 Jul 2023

Genome-wide analysis highlights genetic admixture in exotic germplasm resources of Eucalyptus and unexpected ancestral genomic composition of interspecific hybrids

PONE-D-23-13755R1

Dear Dr. Grattapaglia,

We’re pleased to inform you that your manuscript has been judged scientifically suitable for publication and will be formally accepted for publication once it meets all outstanding technical requirements.

Kind regards,

Chunxian Chen, Ph.D.

Academic Editor

PLOS ONE

---

## [Editor Report · Acceptance letter]

31 Jul 2023

PONE-D-23-13755R1 

Genome-wide analysis highlights genetic admixture in exotic germplasm resources of *Eucalyptus* and unexpected ancestral genomic composition of interspecific hybrids 

Dear Dr. Grattapaglia:

I'm pleased to inform you that your manuscript has been deemed suitable for publication in PLOS ONE. Congratulations! Your manuscript is now with our production department. 

Kind regards, 

on behalf of

Dr. Chunxian Chen 

Academic Editor

PLOS ONE